# Streamline Without Sacrifice - Squeeze out Computation Redundancy in LMM

Penghao Wu [1]  Lewei Lu [2]  Ziwei Liu [1]

## Abstract

Large multimodal models excel in multimodal tasks but face significant computational challenges due to excessive computation on visual tokens. Unlike token reduction methods that focus on token-level redundancy, we identify and study the **computation-level redundancy** on vision tokens to ensure no information loss. Our key insight is that vision tokens from the pretrained vision encoder do not necessarily require all the heavy operations (*e.g.*, self-attention, FFNs) in decoder-only LMMs and could be processed more lightly with proper designs. We designed a series of experiments to discover and progressively squeeze out the vision-related computation redundancy. Based on our findings, we propose **ProxyV**, a novel approach that utilizes proxy vision tokens to alleviate the computational burden on original vision tokens. ProxyV enhances efficiency without compromising performance and can even yield notable performance gains in scenarios with more moderate efficiency improvements. Furthermore, the flexibility of ProxyV is demonstrated through its combination with token reduction methods to boost efficiency further. The code will be made public here.

## 1. Introduction

Large multimodal models (LMMs) have demonstrated powerful capabilities by combining visual information with large language models (LLMs), but their computational overhead can be immense due to the large number of vision tokens. To mitigate this, most existing efforts address **token-level redundancy** by pruning or merging vision tokens with the risk of discarding fine-grained details. In this paper, we instead tackle the **computation-level redundancy** on vision tokens, an often-overlooked dimension. We find that

[1]S-Lab, Nanyang Technological University [2]SenseTime Research. Correspondence to: Penghao Wu <penghao001@e.ntu.edu.sg>, Ziwei Liu <ziwei.liu@ntu.edu.sg>.

*Proceedings of the $42^{nd}$ International Conference on Machine Learning*, Vancouver, Canada. PMLR 267, 2025. Copyright 2025 by the author(s).

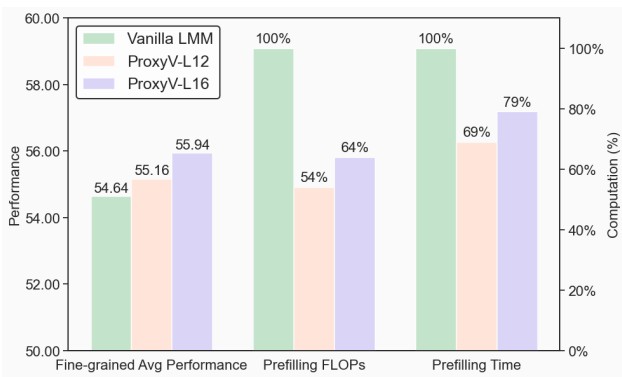

*Figure 1.* ProxyV retains or increases the fine-grained benchmark performance while effectively reducing the computational cost. ProxyV-L12 and ProxyV-L16 denote applying ProxyV from layers 12 and 16, respectively.

computation-level redundancy exists in the processing of vision tokens and can be squeezed out through proper designs without sacrificing performance.

Current mainstream LMMs adopt a typical decoder-only architecture, exemplified by LLaVA-style (Liu et al., 2024b) pipelines, usually comprising: 1) A pretrained vision encoder extracting vision features, 2) A lightweight projection module projecting vision features into LLMs' input space 3) An LLM jointly processing the concatenated vision and text tokens through multiple layers of self-attention and feed-forward networks (FFNs). Though simple and effective, this structure faces a significant challenge: the high computational cost incurred by vision tokens, which typically far outnumber text tokens. Since self-attention scales quadratically with sequence length, the problem becomes even more severe as top-performing LMMs (Li et al., 2024b; Wang et al., 2024; Chen et al., 2024b) usually process high-resolution images, resulting in thousands of tokens per image. Moreover, LMMs are now being extended to process video frames (Lin et al., 2023a; Maaz et al., 2024; Zhang et al., 2024a; Xue et al., 2024) or multiple images (Li et al., 2024c; Jiang et al., 2024a) within the same structure, further increasing the length of visual token sequences.

A popular and intuitive solution to this problem is to reduce the number of vision tokens via pruning and merging

(Chen et al., 2025b; Shang et al., 2024; Zhang et al., 2024b; Xing et al., 2024; Yang et al., 2024b). While effective in certain scenarios, this approach is problematic as it involves non-recoverable operations that risk discarding fine-grained information or contextual details. For instance, in the case of a dense document image, pruning or merging tokens is highly likely to result in the loss of critical information. Several methods (Huang et al., 2025; Xing et al., 2024) use the textual question as guidance for the token pruning or selection process. However, this approach can fail in multi-turn conversations, where later questions might require visual information not retained based on the guidance of the initial question. Additionally, it can be challenging to identify all critical visual information for complex or indirect questions. Furthermore, many token reduction methods rely on text-to-image attention scores, which makes them incompatible with time and memory-efficient attention implementations (Dao et al., 2022; Dao, 2023).

How can we reduce the computation cost brought by long vision sequences while always preserving all vision tokens to avoid any possible information loss? Note that the vision tokens for most LMMs come from a pretrained vision encoder, so the vision tokens are already highly semantic. This raises the question: Is it still necessary to perform all the heavy operations (*e.g.*, vision-to-vision attention and FFNs) on them within the LLM? Is there any computation-level redundancy on vision tokens in LMMs? Cross-attention-based LMMs (Alayrac et al., 2022; Laurencon et al., 2023; Dubey et al., 2024; Dai et al., 2024) already offer some promising insights into these questions. These methods treat vision features as context, injecting them into the LLM via additional cross-attention modules, thereby avoiding unrolling all vision tokens in the LLM and improving computational efficiency. However, these models also have drawbacks such as requiring significantly more pre-training data, bringing a large number of additional parameters, and still slightly under-performing the decoder-only counterparts. The computation-level redundancy on vision tokens is likely to exist, but can we reduce it while retaining the simplicity and efficiency of the decoder-only structure?

Motivated by this, we first conduct experiments to verify the existence of computation-level redundancy. Our findings confirm that attention-related computation redundancy on vision tokens does exist, with different LLMs exhibiting varying degrees of redundancy. We then explore the possibility of skipping both attention and FFN operations on vision tokens by replacing them with lightweight MLP modules. Interestingly, the newly added lightweight modules, which are vision-specific, bring additional performance gains. And the final performance can be understood as the performance gain from the lightweight modules minus the performance drop caused by skipping attention and FFN operations on vision tokens. But can we have a better design to further mit-

igate the negative impact of skipping all heavy operations on vision tokens?

We then propose a better solution, **ProxyV**, that introduces a group of proxy vision tokens to relieve the full vision tokens from the heavy computation burden. As shown in Figure 1, ProxyV achieves 101% performance with prefilling FLOPs and time reduced by 43% and 40% respectively, and achieves 102.4% performance with FLOPs and time reduced by 36% and 33% respectively on benchmarks requiring fine-grained visual understanding when applied to Vicuna1.5-7B. We compare ProxyV against token reduction methods and highlight the information loss problem for them. Furthermore, we explore a non-spatial variant of ProxyV, which can be seamlessly integrated with token reduction methods to further enhance efficiency.

Overall, our contributions are threefold:

- We systematically study the *computation-level redundancy* on vision tokens in decoder-only LMMs and explore ways to progressively reduce it.

- We propose ProxyV, a novel design that introduces proxy tokens to carry out heavy computations, effectively reducing computation while ensuring performance.

- We extensively validate the effectiveness of ProxyV with different LLMs and show its flexibility by proposing a non-spatial variant that can be directly combined with token reduction methods.

## 2. Discover Computation-level Redundancy

To validate our hypothesis regarding computation-level redundancy in decoder-only LMMs, we first design a series of exploratory experiments to investigate the presence of such redundancy in self-attention operations among vision tokens. Specifically, we train a set of LMMs based on the LLaVA-Next (Liu et al., 2024a) structure with different LLM backbones, including Vicuna1.5-7B (Zheng et al., 2023), Vicuna1.5-13B (Zheng et al., 2023), LLama3-8B (Dubey et al., 2024), Qwen2-7B (Yang et al., 2024a), Phi3-3B (Abdin et al., 2024), and InternLM2.5-7B (Cai et al., 2024). Detailed experimental settings are provided in Section 4. During inference, we masked the attention within vision tokens to disable inter-token interactions, applying this attention masking at different positions within the LLM (*i.e.*, across various portions of the decoder layers). To evaluate the impact of this masking on the performance of LMMs, we select a set of OCR-extensive benchmarks (DocVQA (Mathew et al., 2021), ChartQA (Masry et al., 2022), InfoVQA (Mathew et al., 2022), OCRBench (Liu et al., 2024c), TextVQA (Singh et al., 2019)) that require fine-grained visual information, thus being highly sensitive

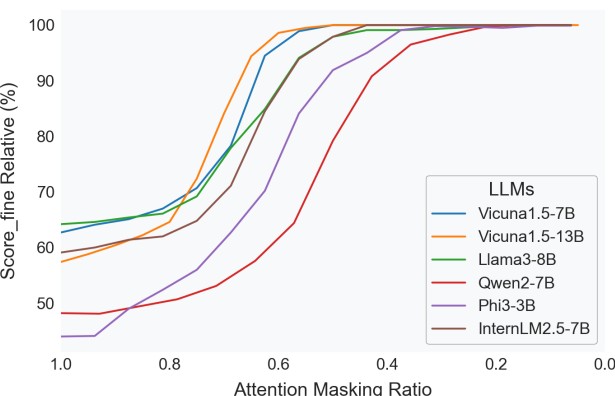

Figure 2. The relative Score_fine with different vision attention masking ratios for different LLMs. The computation redundancy begins in the middle to rear part of the LLMs as masking the vision attention does not affect the performance.

to potential information loss in vision tokens, and denote the average performance on them as $\text{Score}_{\text{fine}}$.

As shown in Figure 2, directly masking vision token attention across the entire LLM leads to a significant performance drop, while masking it from the middle or later layers has minimal or no effect on performance. Furthermore, while the general trend is consistent across all LLMs, different models exhibit different patterns, indicating varying degrees of redundancy. For instance, Vicuna1.5-7B and Vicuna1.5-13B maintain full performance when the masking is applied from the middle layers, while Qwen2-7B achieves 100% performance when only the latter 25% layers are masked. The different patterns of vision attention across LLMs present an intriguing research direction, but we do not explore them in this paper and focus on the general trend. Based on these observations, we conclude that **the attention-related computation redundancy on vision tokens does exist in the middle and later layers of LMMs, with varying degrees of redundancy across different LLMs**.

Since the above experiments are directly conducted in the training-free approach, a natural question arises: *Can we finetune the model with the vision-to-all attention operations skipped to further reduce the performance gap?* In practice, skipping the vision-to-all attention operation means that the vision tokens only act as the keys and values in the attention operation and are updated directly with v-projection and o-projection, which is also the implementation studied in EE-MLLM (Ma et al., 2024). We then conduct this experiment with the Vicuna1.5-7B LLM and skip the vision-to-all attention from layers 0, 12, and 16, respectively. As shown in Table 1, **finetuning with the vision-to-all attention skipped mitigates the performance drop**. We also report the reduced FLOPs and time cost for the prefilling

stage in Table 1. Notably, even when vision attention is skipped across all layers, the reduction in FLOPs is limited. This is primarily due to the computationally intensive nature of the heavy FFN operations on vision tokens. This leads to another question: *is it possible to also skip the heavy FFNs or replace them with lightweight alternatives?*

Table 1. Results of finetuning LMMs with vision-to-all attention skipped. TF denotes the training-free approach by masking vision attention during inference and FT denotes finetuning with vision attention skipped. L0, L12, and L16 indicate the layer index where the masking or skipping starts to be applied. Finetuning can further improve the performance, but FLOPs reduction is limited.

|  | $\text{Score}_{\text{fine}}$ Rel | FLOPs | Time |
|---|---|---|---|
| L0 - TF | 62.7% | - | - |
| L0 - FT | 87.2% | ↓ 18% | ↓ 60% |
| L12 - TF | 94.5% | - | - |
| L12 - FT | 99.3% | ↓ 11% | ↓ 40% |
| L16 - TF | 100.2% | - | - |
| L16 - FT | 99.9% | ↓ 9% | ↓ 32% |

Table 2. Results of replacing the original attentions and FFNs on vision tokens with lightweight MLPs. ATN represents the previous approach that only skips the attention, while ATN-FFN represents skipping both attention and FFNs. Skipping FFNs reduces the FLOPs, and adding lightweight MLPs brings a performance gain.

|  | $\text{Score}_{\text{fine}}$ Rel | FLOPs | Time |
|---|---|---|---|
| L0 - ATN | 87.2% | ↓ 18% | ↓ 60% |
| L0 - ATN+FFN | 65.1% | ↓ 80% | ↓ 77% |
| L12 - ATN | 99.3% | ↓ 11% | ↓ 40% |
| L12 - ATN+FFN | 99.3% | ↓ 50% | ↓ 49% |
| L16 - ATN | 99.9% | ↓ 9% | ↓ 32% |
| L16 - ATN+FFN | 100.4% | ↓ 40% | ↓ 39% |

Our initial attempt reveals that directly skipping FFNs and keeping vision tokens constant across decoder layers severely degrades the performance, so we use lightweight MLPs to replace the attention operations and FFNs for the vision token update. The corresponding results, shown in Table 2, indicate that this approach significantly reduces FLOPs and further improves the speed. Compared to skipping attention operations alone, additionally skipping the FFNs degrades the performance for the layer-0 case, achieves similar performance for the layer-12 case, and even improves the performance for the layer-16 case. This interesting performance improvement arises because the addition of lightweight MLPs introduces decoupled vision-specific modules, which benefit overall performance. This finding also partially aligns with findings from Libra (Xu

et al., 2024), which demonstrated that the decoupled vision-specific modeling can better process the vision-specific information without distorting the original knowledge in the LLM. However, unlike Libra, which employs a substantial number of vision-specific parameters, we achieve this with lightweight MLPs (9.44M parameters per layer). The final performance can thus be understood as **the performance gain from the newly added vision-specific parameters** minus **the performance drop caused by skipping the original heavy operations on vision tokens**. Based on these observations, instead of this simple implementation, *can we have a better design to eliminate performance loss further or even enhance performance while maintaining computational efficiency?*

## 3. A Better Solution with Proxy Vision Tokens

We now introduce our algorithm **ProxyV**, with its overall framework shown in Figure 3. The core idea of ProxyV is to employ a small group of proxy vision tokens as substitutes for the original vision tokens in compute-intensive operations. These proxy tokens then guide the updates of the original vision tokens through lightweight modules.

### 3.1. Proxy Tokens with Spatial Priors

As the original full vision tokens still preserve the 2D spatial structure, a direct approach is to downsample them into a smaller vision feature map to serve as proxy tokens. Specifically, given $N \times N$ original vision tokens, we downsample them by a factor $r$ to get a thumbnail version of size $M \times M$ where $M = N/r$. In the LLM decoder layer, the proxy vision tokens and text tokens serve as queries, while the values and keys consist of the proxy vision tokens, original full vision tokens, and text tokens. After the attention operation, only the proxy tokens and text tokens are processed by the FFNs. In this way, proxy vision tokens replace full vision tokens in the compute-intensive operations, significantly reducing computational cost. After the proxy tokens obtain useful information through these operations, each proxy token guides the spatially corresponding $r \times r$ full vision tokens for an update through a lightweight guided-update module. Through this design, the important information in the heavy computation of the decoder layer could be effectively obtained and transferred to the full vision tokens with proxy vision tokens as an intermediary. As a result, the negative effect of skipping all extensive operations is effectively mitigated without sacrificing much efficiency.

In our implementation of the guided-update module, we first down-project the full and proxy vision tokens with linear layers and then directly concatenate the full vision tokens with their spatially corresponding proxy vision tokens and process them with a lightweight two-layer MLP to update the full vision tokens. Note that this guided-update module

could also include some local attention layer or convolution layer to further promote the fine-grained inter-token interactions in each local $r \times r$ window, and we leave this for future work. We validate the ProxyV design in the same setting as Section 2 and the results are shown in Table 3. With ProxyV, the negative effect of skipping attention and FFNs on full vision tokens is effectively mitigated, and now the performance gain brought by decoupled vision-specific modules becomes more pronounced, leading to better overall performance at the layer 12 and 16 cases.

*Table 3.* Results of applying ProxyV from different layers. ProxyV effectively mitigates the performance drop caused by skipping operations on vision tokens and brings additional performance gain with vision-specific modules.

|  | Score$_{\text{fine}}$ Rel | FLOPs | Time |
|---|---|---|---|
| L0 - ProxyV | 88.9% | ↓ 73% | ↓ 68% |
| L12 - ProxyV | 101.0% | ↓ 46% | ↓ 41% |
| L16 - ProxyV | 102.4% | ↓ 36% | ↓ 31% |

We further validate the effectiveness of our ProxyV algorithms with different LLM backbones in Table 4. The results indicate that applying ProxyV from the middle layers can achieve no performance loss or a small performance gain (100% - 101%) with moderate efficiency improvement. Applying it from the middle and rear part of the LLM achieves notable performance improvement (101% - 102%) with a smaller efficiency gain. We provide the evaluation results on additional general LMM benchmarks in the Supplementary.

We also conduct a preliminary study to reveal the possible internal mechanisms underlying the performance improvement. Specifically, we measured the MIR (Modality Integration Rate) score (Huang et al., 2024), which quantifies the degree of alignment between visual and textual tokens within LMMs (lower scores indicate better alignment). We randomly sample 100 instances from the dense captioning dataset DetailCaps-4870 (Dong et al., 2024) and measure the MIR scores of the baseline and the variant where the vision-specific MLPs replace the original vision operations from layer 0. ProxyV with vision-specific MLPs reduces the MIR scores from 3.62 to 3.10. This indicates that the vision-specific parameters achieve better alignment between text and vision tokens, leading to performance improvement. We leave further study about this direction as future work.

### 3.2. Comparison with Token Reduction Methods

Here we compare ProxyV with two state-of-the-art token reduction methods: VisionZip (Yang et al., 2024b) and PyramidDrop (Xing et al., 2024) with the Vicuna1.5-7B LLM setting. VisionZip performs token reduction before the LLM by selecting a group of dominant vision tokens and merging

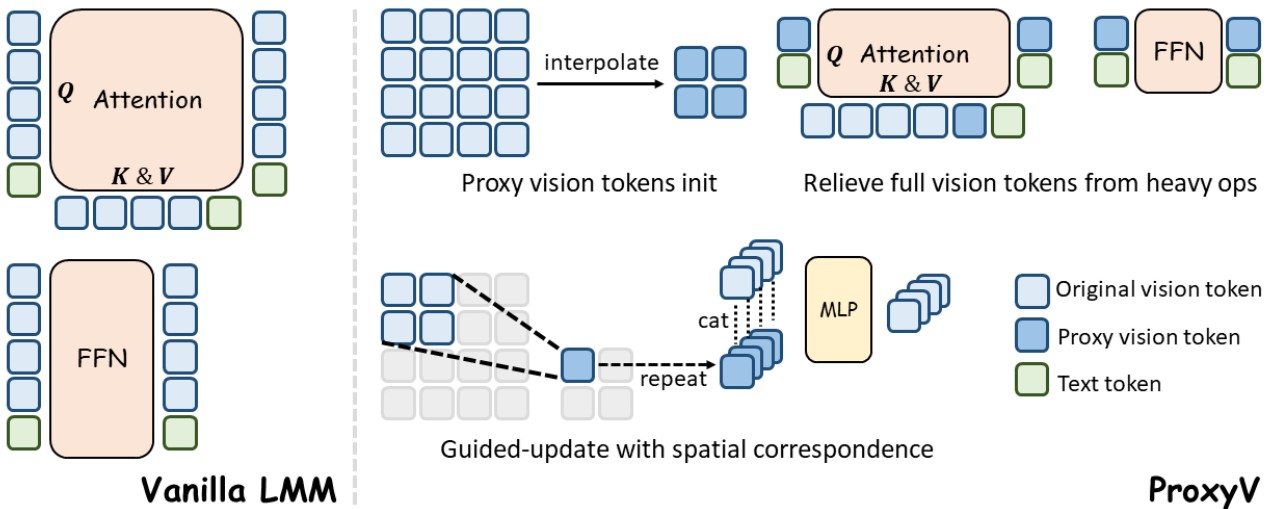

*Figure 3.* Left: the vanilla LMM structure where full vision tokens cause significant computation. Right: the overall pipeline of the proposed ProxyV algorithm. The full vision tokens are first downsampled to obtain a much smaller version that works as proxy vision tokens. The proxy vision tokens participate in the original operations in the decoder layer including the self-attention and the FFNs to obtain useful information at a much lower cost. After this, each original vision token is guided by its spatially corresponding proxy vision token for an update through a lightweight MLP.

the remaining as contextual tokens. PyramidDrop progressively reduces the number of vision tokens inside the LLM, utilizing the attention scores. We configure these two methods so that the efficiency improvement is similar to ours, and finetune the LMMs for fair comparison. We acknowledge that token-level redundancy does exist in general scenarios, but our goal is to ensure no vision information loss for all cases even when the images contain very dense visual information or require accurate visual grounding. Therefore, we also evaluate the models on a grounding benchmark Ref-COCO (Kazemzadeh et al., 2014) and additionally include the document parsing task to simulate this scenario. For the document parsing task, we continue to train the models on the 1M document parsing data from DocStruct4M (Hu et al., 2024) dataset and evaluate them on the CCpdf (Turski et al., 2023) dateset in the validation split. We measure the BLEU (Papineni et al., 2002) and edit distance for this task.

As shown in Table 5, VisionZip and PyramidDrop achieve nearly no performance drop on selected benchmarks but have notable degradation and grounding benchmark and much worse performance on the document parsing task, highlighting the issue of visual information loss inherent to token reduction methods. We also provide qualitative examples in Figure 5 to show the failure of token reduction methods where dense or structured visual information needs to be extracted or the image contains dense information and visual details, while ProxyV retains all visual information.

### 3.3. A ProxyV Variant without Spatial Constraints

Though we directly compare ProxyV with token reduction methods in the previous section, our goal is to diminish computation-level redundancy, which is theoretically orthogonal to the objective of token reduction methods that focus on removing token-level redundancy. This raises the question: *Is it possible to combine ProxyV with these token reduction methods?* The primary challenge in directly combining the two approaches lies in the fact that ProxyV relies on the 2D spatial structure of vision tokens for generating proxy tokens and establishing correspondence in the guided-update module. However, after applying token reduction methods, the spatial structure of the vision tokens is no longer preserved, making integration nontrivial.

To resolve this problem, we propose a non-spatial variant of the original ProxyV algorithm to remove the requirement of a spatial prior so that this alternative can be flexibly combined with token reduction methods or non-spatial vision features. Specifically, we initialize a set of learnable embeddings $Q \in \mathbb{R}^{m \times d}$ where $m$ is the number of desired proxy tokens and $d$ is a hidden dimension typically set smaller than the hidden dimension of the LLM. A linear layer projects the full vision tokens to $K \in \mathbb{R}^{n \times d}$ where $n$ is the total number of full vision tokens. And we also define values $V \in \mathbb{R}^{n \times d_{hidden}}$ to be directly equal to the full vision tokens, where $h_{hidden}$ is the dimension of the full vision tokens.

A vanilla attention operation is then applied on $Q, K$, and

*Table 4.* The results of applying ProxyV on different LLMs. Applying ProxyV from the middle layers ensures no performance drop, and applying it from the middle to rear layers achieves notable performance improvement.

| | Score$_{fine}$ | Score$_{fine}$ Relative | FLOPs reduced | Time reduced |
|---|---|---|---|---|
| Vicuna-7B | | | | |
| Baseline | 54.64 | 100.0% | – | – |
| ProxyV - Layer 12 | 55.16 | 101.0% | 46% | 41% |
| ProxyV - Layer 16 | **55.94** | **102.4%** | 36% | 31% |
| Vicuna-13B | | | | |
| Baseline | 58.58 | 100.0% | – | – |
| ProxyV - Layer 16 | 58.95 | 100.6% | 43% | 40% |
| ProxyV - Layer 20 | **59.22** | **101.1%** | 36% | 33% |
| Llama3-8B | | | | |
| Baseline | 56.60 | 100.0% | – | – |
| ProxyV - Layer 16 | 56.87 | 100.5% | 42% | 34% |
| ProxyV - Layer 20 | **57.49** | **101.6%** | 32% | 25% |
| Qwen2-7B | | | | |
| Baseline | 60.15 | 100.0% | – | – |
| ProxyV - Layer 16 | 60.55 | 100.7% | 37% | 29% |
| ProxyV - Layer 20 | **61.44** | **102.1%** | 25% | 14% |
| Phi3-3B | | | | |
| Baseline | 49.89 | 100.0% | – | – |
| ProxyV - Layer 16 | 50.28 | 100.8% | 35% | 34% |
| ProxyV - Layer 20 | **50.72** | **101.7%** | 27% | 25% |
| InternLM2.5-7B | | | | |
| Baseline | 58.33 | 100.0% | – | – |
| ProxyV - Layer 16 | 58.68 | 100.6% | 39% | 33% |
| ProxyV - Layer 20 | **59.08** | **101.3%** | 30% | 24% |

$V$ to get the attention logits (before softmax) $A \in \mathbb{R}^{m \times n}$. As a result, the proxy tokens are the direct weighted combination of the full vision tokens as $softmax(A, dim = -1)V$. During the guided update process for the full vision tokens, the previous ProxyV algorithm with spatial constraints mapped each full vision token to a proxy token based on spatial correspondence. In this non-spatial variant, we re-use the attention logits matrix $A$, transpose it, apply the softmax along the proxy token dimension, and multiply it with the proxy tokens to get the guidance for all full vision tokens as the weighted combination of proxy tokens. The process is illustrated in Figure 4.

First, we validate the feasibility of this non-spatial variant version of ProxyV. Subsequently, we combine it with VisionZip to explore the possibility of combining our approach with token reduction methods. As shown in Table 6, the non-spatial ProxyV variant attains a similar performance as the original one, and combining it with VisioZip achieves the desired performance with further increased efficiency.

## 4. Experiment Details

For all experiments, we use the widely adopted 2-stage training pipeline. For stage 1, we pretrain the multi-modal projector and the newly added vision-specific modules using 1.2M captioning data from ShareGPT4V (Chen et al., 2025a) for 1 epoch. For the finetuning stage, we train the model for 1 epoch using the 779K instruction tuning data in LLava-Next (Liu et al., 2024a) and unfreeze the LLM in this stage.

For image encoding, we adopt the AnyRes strategy (Liu et al., 2024a) with a maximal 5 grids per image, including the thumbnail one. Each grid with resolution 336×336 is encoded by `CLIP-ViT-L-336px` (Radford et al., 2021) to a 24×24 image feature. The image feature is further projected by a 2-layer MLP projector and flattened in raster order within each grid, and concatenated grid by grid, similar to the UniRes strategy (Zhang et al., 2024a). We also append one separator token after each grid.

For our ProxyV implementation, we choose the downsampling factor $r = 4$ so that 576 full vision tokens are com-

*Table 5.* Comparison between ProxyV and token reduction methods. Token reduction methods significantly underperform on the document parsing task, indicating the inherent information loss problem.

| Methods | $\text{Score}_{\text{fine}}$ | RefCOCO | DocParsing-BLEU ↑ | DocParsing-ED ↓ | FLOPs | Time |
|---|---|---|---|---|---|---|
| Baseline | 54.64 | 82.96 | 0.7991 | 0.1551 | – | – |
| ProxyV - Layer 12 | 55.16 | 84.41 | 0.7923 | 0.1566 | ↓ 46% | ↓ 41% |
| VisionZip | 54.58 | 79.11 | 0.7218 | 0.1734 | ↓ 32% | ↓ 40% |
| PyramidDrop | 54.63 | 80.75 | 0.7261 | 0.1702 | ↓ 42% | ↓ 46% |

*Table 6.* The results of the non-spatial ProxyV variant (the ProxyV-ns entry) and the combination of non-spatial ProxyV and VisionZip (the Combination entry). The non-spatial version achieves similar performance and combining it with token reduction methods further boosts the efficiency while maintaining the performance.

| | $\text{Score}_{\text{fine}}$ | FLOPs | Time |
|---|---|---|---|
| Baseline | 54.64 | – | – |
| VisionZip | 54.58 | ↓ 32% | ↓ 40% |
| ProxyV - Layer12 | 55.16 | ↓ 46% | ↓ 41% |
| ProxyV-ns - Layer12 | 54.85 | ↓ 44% | ↓ 44% |
| Combination | 54.83 | ↓ 62% | ↓ 65% |

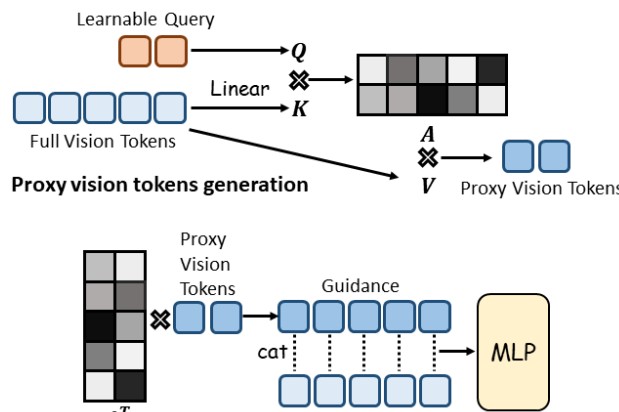

*Figure 4.* The illustration of the non-spatial ProxyV. Upper part: proxy vision tokens are generated as a weighted combination of full vision tokens through a simple attention operation. Lower part: The previous attention score is reused to splat the proxy vision tokens into guidance for the full vision tokens update. The softmax operations are skipped in the figure.

pressed to 36 proxy vision tokens, and each proxy token corresponds to 16 full vision tokens in the guided-update process. For the non-spatial ProxyV version, we set the number of learnable queries to be the same as the spatial version. The hidden dimension in the guided-updated MLP module is set to be $\frac{1}{4}$ of the hidden dimension in the LLM. The number of parameters of the newly added guided-update module for each layer is 14.68M for the Vicuna1.5-7B case. For the VisionZip baseline, we use 360 dominant tokens and 40 contextual tokens. For the PyramidDrop baseline, the vision token is reduced by 50% after layers 12, 20, and 26.

The reported FLOPs and time for all experiments are measured during the prefilling stage, using a fixed configuration of five image grids (2880 tokens) and 50 text tokens, with eager attention implementation on a single H100 GPU. The full evaluation results are provided in Appendix B.

## 5. Related Works

### 5.1. Large Multimodal Models

As large language models achieved unprecedented success, large multimodal models (Liu et al., 2024b; Dai et al., 2023; Liu et al., 2024a; Lin et al., 2023b; Tong et al., 2024; Li et al., 2024b; Wang et al., 2024; Chen et al., 2024b) have emerged, leveraging LLMs as a foundation while incorporating multimodal capabilities by injecting multimodal information extracted from pretrained multimodal encoders

into the LLMs.

Current LMMs can be broadly categorized into two groups: decoder-only LMMs and cross-attention-based LMMs. Decoder-only LMMs (Liu et al., 2024b; Wang et al., 2024; Chen et al., 2024b) directly concatenate the projected visual tokens with textual tokens before LLM and process them equally as text tokens in the LLM through self-attention decoder layers. This approach has validated its simplicity and effectiveness, but incurs high computational costs with long vision sequences. Cross-attention based LMMs (Alayrac et al., 2022; Laurencon et al., 2023; Dubey et al., 2024; Dai et al., 2024), on the contrary, treat the visual information as context and introduce additional cross-attention layers to let text tokens extract visual information. These methods avoid the computationally intensive vision-to-vision attention. But they also bring additional training complexity and a non-negligible amount of parameters compared with decoder-only ones and usually require a significantly larger amount of data for pertaining cross-attention mod-

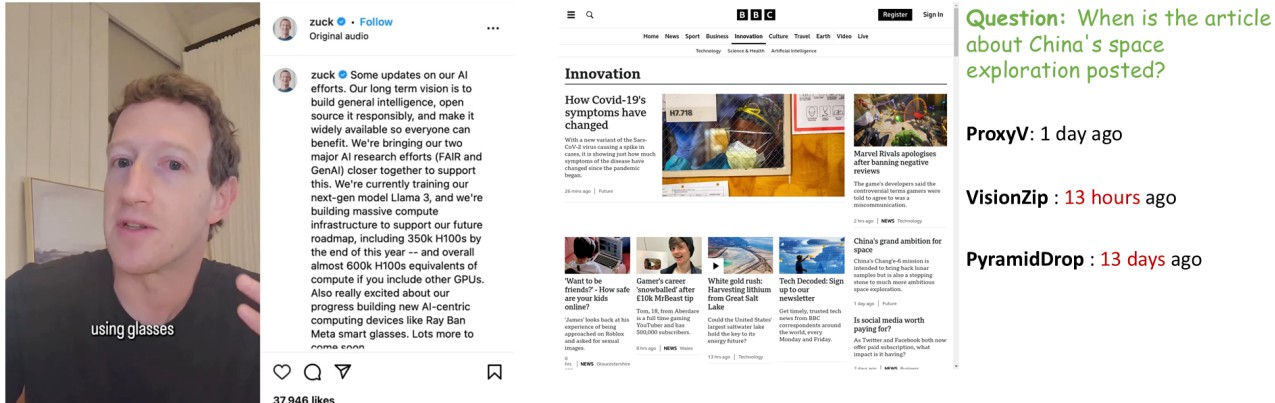

**Question:** What is the exact content of the tweet?

**ProxyV**: Some updates on our AI efforts. Our long term vision is to build general intelligence, open source it responsibly, and make it widely available so everyone can benefit. We're bringing our two major AI research efforts (FAIR and GenAI) closer together to support this. We're currently training our next-gen model Llama 3, and we're building massive compute infrastructure to support our future roadmap, including 350k H100s by the end of this year — and almost 600k H100s equivalents of compute if you include other GPUs. Also really excited about our progress building new AI-centric computing devices like Ray Ban Meta smart glasses. Lots more to come soon.

**VisionZip**: Some updates on our Al efforts. Our long term vision is to build general intelligence, open source it responsibly, and make it widely available so everyone can benefit. We're bringing our two major Al research efforts (FAIR and GenAI) closer together to support our future roadmap. Including 350k H100s by the end of this year — and overall almost 600k H100s equivalents of compute if you include other GPUs. Also really excited about our progress building new Al-Centric computing devices like Ray Ban Meta smart glasses. Lots more to come soon.

**PyramidDrop**: Some updates on Al. Our long term vision is to build general intelligence, open source it, make it readily available so everyone can benefit. We're bringing our two- GenAI closer together to support our future roadmap, including 350k H100s by the end of this year — and overall almost 600k H100s equivalents of compute if you include other GPUs. Also really excited about our progress building new AI-centric computing devices like Ray Ban Meta smart glasses. Lots more to come soon.

*Figure 5.* Cases where token reduction methods fail. Left: Token reduction methods fail to extract the complete dense information accurately. Right: Token reduction methods fail to retain critical visual information when the image contains diverse and dense visual details. In these cases, ProxyV retains all the visual information and successfully extracts the important visual details.

ules. Besides, cross-attention based methods still slightly underperform the decoder-only ones, especially on tasks requiring fine-grained visual information.

Early LMMs typically processed low-resolution images (e.g., $336 \times 336$), which significantly constrained their performance due to the limited input resolution. Recent advancements in LMMs have adopted multi-grid image encoding schemes (Liu et al., 2024a) or directly processed native high-resolution images (Wang et al., 2024), leading to substantial performance improvements and unlocking their potential for a wide range of applications. However, this capability comes at a cost, as the computational burden increases sharply due to the quadratic scaling of attention computation with the number of vision tokens. This challenge becomes even more pronounced with video LMMs (Lin et al., 2023a; Maaz et al., 2024; Zhang et al., 2024a;

Xue et al., 2024) and multi-image LMMs (Li et al., 2024c; Jiang et al., 2024a), which introduce significantly more vision tokens, further exacerbating the computational load.

### 5.2. Token Reduction in LMMs

As LMMs face significant computational costs, especially with long vision sequences, extensive research has focused on reducing these costs through vision token reduction. The stages at which vision token reduction occurs can be grouped into three categories: (1) before the LLM, (2) during the prefilling stage, and (3) during the decoding stage.

**Token Reduction Before the LLM:** Methods in this category directly reduce the number of vision tokens either within the vision encoder or from its outputs, utilizing the attention scores in the vision encoders (Shang et al., 2024;

Jiang et al., 2024b; Yang et al., 2024b). LLaVA-Prumerge (Shang et al., 2024) utilizes the attention scores between CLS token and other vision tokens from the vision encoder to choose crucial tokens to retain which are then further grouped and merged. FoPru (Jiang et al., 2024b) adopts two strategies to calculate the significance of vision tokens in global and local views respectively. VisionZip (Yang et al., 2024b) also uses the vision attention scores to select informative tokens but they merge the remaining tokens to reduce information loss.

**Token Reduction During the Prefilling Stage:** Methods that conduct token reduction during the prefilling stage within the LLM often use the attention scores in the LLM self-attention for token reduction (Chen et al., 2025b; He et al., 2024; Zhang et al., 2024b; Xing et al., 2024; Endo et al., 2024; Zhong et al., 2024). Some methods (He et al., 2024; Zhang et al., 2024b; Xing et al., 2024; Zhong et al., 2024) also adopt adaptive strategies to progressively prune or merge the vision tokens along the layers.

FastV (Chen et al., 2025b) uses the attention patterns in early layers to prune vision tokens in the subsequent layers. ZipVL(He et al., 2024) designs a dynamic ratio allocation strategy to adaptively determine important vision tokens layer-specific attention statistics. SparseVLM (Zhang et al., 2024b) selects visual-relevant text tokens to rate the significance of vision tokens using the LLM self-attention scores and progressively prunes the vision tokens with adaptive scarification ratios across layers. PyramidDrop (Xing et al., 2024) uses the text-to-vision attention scores to progressively prune vision tokens in multiple stages across layers. G-search and P-Sigmoid are proposed in (Zhao et al., 2024) to search for the optimal reduction ratio per layer. FEATHER (Endo et al., 2024) identifies the positional bias problem in previous methods and proposes to incorporate uniform sampling to ensure uniform coverage. AIM (Zhong et al., 2024) first merges the vision tokens before LLM based on embedding similarity and then adopts the page rank algorithm to progressively prune vision tokens in the LLM.

**KV-Cache Compression in Decoding:** KV-Cache compression during the decoding stage is widely studied in the LLM field (Xiao et al., 2023; Ge et al., 2023; Zhang et al., 2023). For LMMs, LOOK-M (Wan et al., 2024) designs a text-prior KV pairs eviction strategy for multimodal KV cache pruning and provides different strategies to merge KV pairs. ElasticCache (Liu et al., 2024d) uses important key/value vectors as anchors and merges the less important ones into them. CSP (Pei et al., 2024) achieves more precise KV cache pruning by independently managing the inter-modality and intra-modality attention.

Token reduction methods effectively remove token-level redundancy when it exists in the image in certain scenarios.

However, the reduction also inevitably introduces information loss when dense information is contained in the image and little redundancy exists. And the instruction/text guided pruning method does not consider the multi-turn conversation cases where later questions might require different vision information. Also, for complex or indirect questions, it is hard to accurately select all the critical vision tokens. On the contrary, our method takes another perspective to reduce the computation-level redundancy to avoid information loss and can also be flexibly combined with these token reduction methods.

## 6. Conclusion

In this paper, we reveal the computation-level redundancy with vision tokens in LMMs. We further explore gradually skipping the heavy attention and FFNs operations and find that using lightweight vision-specific MLPs as a replacement is able to compensate for the performance drop. We then propose a better solution ProxyV which utilizes proxy vision tokens to reduce the negative influence brought by skipping heavy operations on full vision tokens. Experiments on different LLMs validate the effectiveness of ProxyV. We also design a non-spatial variant of ProxyV which can be seamlessly combined with token reduction methods for better efficiency.

## Acknowledgments

This study is supported by the Ministry of Education, Singapore, under its MOE AcRF Tier 2 (MOE-T2EP20221-0012, MOE-T2EP20223-0002), and under the RIE2020 Industry Alignment Fund – Industry Collaboration Projects (IAF-ICP) Funding Initiative, as well as cash and in-kind contribution from the industry partner(s).

## Impact Statement

This work aims to address the computational challenges of LMMs while maintaining or enhancing their performance. The proposed approach reduces computational costs, making LMMs more accessible and environmentally sustainable by decreasing energy consumption. However, like other LMMs and LLMs, ProxyV may inherit ethical considerations related to bias, misuse, or privacy concerns depending on the datasets and tasks it is applied to. Utilization of ProxyV should focus on these aspects to ensure responsible and equitable deployment of such technologies.

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

## A. Evaluation on More Benchmarks

Besides the fine-grained benchmarks, here we evaluate models on a wide range of general multimodal benchmarks including MMBench (Liu et al., 2025), SEED-Bench (Li et al., 2023a), RefCOCO (Kazemzadeh et al., 2014), MMStar (Chen et al., 2024a), GQA (Hudson & Manning, 2019), MME (Fu et al., 2023), MMMU (Yue et al., 2024), POPE (Li et al., 2023b), ScienceQA (Lu et al., 2022), AI2D (Kembhavi et al., 2016), and RealWorldQA (xAI, 2024). We use the en-dev split of MMBench, the image-split of SEED-Bench, the test-A and test-B splits of RefCOCO, the testdev-balanced split of GQA, the perception split of MME, the validation split of MMMU, and the image split of ScienceQA.

We observe that on such general benchmarks where less fine-grained visual information is needed, the redundancy of visual information appears in earlier layers and the performance loss from skipping operations is smaller. Consequently, applying ProxyV from middle-to-later layers achieves similar or better overall performance and better efficiency than later layers, as it introduces vision-specific modules in more layers, leading to a larger gain. We provide results of applying ProxyV on different LLMs together with the two token reduction baselines in Table 7. We can observe that ProxyV consistently achieves no performance loss or even improvements on general multimodal benchmarks.

*Table 7.* Evaluation on a comprehensive set of general multimodal benchmarks.

| | MMBench | SEED-Img | RefCOCO | MMStar | GQA | MME-P | MMMU | POPE | SQA | AI2D | RealWorldQA |
|---|---|---|---|---|---|---|---|---|---|---|---|
| Vicuna-7B | | | | | | | | | | | |
| Baseline | 65.03 | 68.60 | 75.39 | 37.70 | 63.36 | 1428.52 | 36.56 | 86.91 | 68.52 | 66.71 | 56.73 |
| ProxyV-L12 | 67.61 | 70.02 | 76.76 | 38.35 | 64.02 | 1478.61 | 35.44 | 86.55 | 68.07 | 69.11 | 59.08 |
| VisionZip | 65.37 | 68.74 | 71.63 | 37.38 | 63.93 | 1437.50 | 34.33 | 87.44 | 69.51 | 68.30 | 57.52 |
| PyramidDrop | 65.37 | 68.45 | 72.53 | 36.66 | 63.84 | 1451.33 | 35.56 | 87.10 | 67.63 | 67.94 | 58.04 |
| Vicuna-13B | | | | | | | | | | | |
| Baseline | 67.78 | 70.45 | 81.68 | 41.6 | 65.19 | 1617.57 | 34.56 | 86.58 | 71.05 | 69.88 | 58.95 |
| ProxyV-L16 | 69.15 | 71.71 | 83.30 | 42.40 | 65.59 | 1602.15 | 36.00 | 87.16 | 73.53 | 71.50 | 58.82 |
| Llama3-8B | | | | | | | | | | | |
| Baseline | 70.01 | 72.11 | 71.8 | 44.04 | 64.63 | 1470.08 | 37.56 | 87.81 | 77.64 | 72.54 | 59.61 |
| ProxyV-L16 | 70.53 | 72.44 | 73.32 | 44.61 | 64.28 | 1480.87 | 36.67 | 87.17 | 74.22 | 72.93 | 59.74 |
| Qwen2-7B | | | | | | | | | | | |
| Baseline | 73.71 | 72.62 | 84.28 | 48.73 | 64.77 | 1579.88 | 42.67 | 87.19 | 77.59 | 74.13 | 62.61 |
| ProxyV-L16 | 75.51 | 73.58 | 90.37 | 51.38 | 64.28 | 1543.9 | 44.67 | 87.5 | 77.24 | 74.94 | 63.53 |
| Phi3-3B | | | | | | | | | | | |
| Baseline | 64.69 | 68.16 | 52.63 | 38.36 | 61.27 | 1477.93 | 38.56 | 85.57 | 70.10 | 67.68 | 56.73 |
| ProxyV-L16 | 67.95 | 68.77 | 56.22 | 40.60 | 61.64 | 1437.92 | 39.78 | 85.76 | 70.85 | 68.17 | 57.52 |
| InternLM2.5-7B | | | | | | | | | | | |
| Baseline | 74.14 | 74.55 | 66.36 | 48.07 | 64.52 | 1455.53 | 42.56 | 87.78 | 77.24 | 75.61 | 63.52 |
| ProxyV-L16 | 76.20 | 75.07 | 77.4 | 49.52 | 65.39 | 1465.65 | 43.00 | 87.13 | 77.98 | 74.22 | 65.35 |

## B. Fine-grained Benchmark Details and Full Results

For all evaluations, we use the validation splits of DocVQA (Mathew et al., 2021), InfoVQA (Mathew et al., 2022), and TextVQA (Singh et al., 2019). We use the English dev split for MMBench (Liu et al., 2025) and the perception split for MME (Fu et al., 2023) (the score is normalized to 0-100 when calculating $Score_{coarse}$). For the grounding benchmark RefCOCO, we calculate the average of the testA and testB splits. All the evaluations are conducted using the lmms-eval framework (Li et al., 2024a).

The full benchmark results of Tables 1 to 3 are shown in Table 8 and the full results of Table 4 are provided in Table 10.

*Table 8.* Full results of the explorative experiments and ProxyV on the fine-grained benchmarks.

|                | DocVQA | ChartQA | InfoVQA | TextVQA | OCRBench | $Score_{fine}$ |
|----------------|--------|---------|---------|---------|----------|----------------|
| Baseline       | 68.03  | 59.64   | 33.60   | 62.12   | 49.80    | 54.64          |
| L0 - TF        | 35.82  | 31.92   | 25.74   | 45.85   | 31.90    | 34.25          |
| L0 - FT        | 57.50  | 48.48   | 31.38   | 58.29   | 42.70    | 47.67          |
| L0 - ATN+FFN   | 40.19  | 36.60   | 25.60   | 45.88   | 29.60    | 35.57          |
| L0 - ProxyV    | 57.99  | 51.24   | 30.88   | 59.07   | 43.70    | 48.58          |
| L12 - TF       | 63.20  | 55.40   | 31.59   | 60.53   | 47.40    | 51.62          |
| L12 - FT       | 66.55  | 58.48   | 34.27   | 61.36   | 50.70    | 54.27          |
| L12 - ATN+FFN  | 67.45  | 59.48   | 34.77   | 60.72   | 49.00    | 54.28          |
| L12 - ProxyV   | 68.18  | 60.16   | 34.77   | 61.69   | 51.00    | 55.16          |
| L16 - TF       | 68.46  | 59.20   | 33.55   | 61.80   | 50.70    | 54.74          |
| L16 - FT       | 68.09  | 59.40   | 33.35   | 62.07   | 50.00    | 54.58          |
| L16 - ATN+FFN  | 68.95  | 59.32   | 33.49   | 61.29   | 51.20    | 54.85          |
| L16 - ProxyV   | 69.90  | 61.48   | 34.24   | 62.28   | 51.80    | 55.94          |

*Table 9.* Full results of the token reduction methods, the non-spatial ProxyV, and the combination of Proxvy-ns and VisionZip.

|                      | DocVQA | ChartQA | InfoVQA | TextVQA | OCRBench | $Score_{fine}$ |
|----------------------|--------|---------|---------|---------|----------|----------------|
| Baseline             | 68.03  | 59.64   | 33.60   | 62.12   | 49.80    | 54.64          |
| ProxyV - Layer12     | 68.18  | 60.16   | 34.77   | 61.69   | 51.00    | 55.16          |
| VisionZip            | 68.84  | 58.04   | 33.35   | 62.26   | 50.40    | 54.58          |
| PyramidDrop          | 68.50  | 58.88   | 34.43   | 61.84   | 49.50    | 54.63          |
| ProxyV-ns - Layer12  | 68.16  | 60.28   | 34.27   | 61.56   | 50.00    | 54.85          |
| ProxyV-ns + VisionZip| 68.31  | 58.76   | 34.17   | 62.29   | 50.60    | 54.83          |

*Table 10.* Full results of ProxyV with different LLMs on the fine-grained benchmarks.

| | DocVQA | ChartQA | InfoVQA | TextVQA | OCRBench | $\text{Score}_{\text{fine}}$ |
|---|---|---|---|---|---|---|
| Vicuna-7B | | | | | | |
| Baseline | 68.03 | 59.64 | 33.60 | 62.12 | 49.80 | 54.64 |
| ProxyV - Layer 12 | 68.18 | 60.16 | 34.77 | 61.69 | 51.00 | 55.16 |
| ProxyV - Layer 16 | 69.90 | 61.48 | 34.24 | 62.28 | 51.80 | 55.94 |
| Vicuna-13B | | | | | | |
| Baseline | 72.11 | 63.64 | 37.52 | 65.35 | 54.30 | 58.58 |
| ProxyV - Layer 16 | 73.69 | 65.04 | 38.09 | 64.72 | 53.20 | 58.95 |
| ProxyV - Layer 20 | 73.83 | 65.32 | 37.52 | 65.65 | 53.80 | 59.22 |
| Llama3-8B | | | | | | |
| Baseline | 71.87 | 63.64 | 32.20 | 63.08 | 52.20 | 56.60 |
| ProxyV - Layer 16 | 73.08 | 63.00 | 33.81 | 62.86 | 51.60 | 56.87 |
| ProxyV - Layer 20 | 72.89 | 63.32 | 35.09 | 62.84 | 53.30 | 57.49 |
| Qwen2-7B | | | | | | |
| Baseline | 76.59 | 65.76 | 42.31 | 63.69 | 52.40 | 60.15 |
| ProxyV - Layer 16 | 76.54 | 65.80 | 42.31 | 63.99 | 54.10 | 60.55 |
| ProxyV - Layer 20 | 78.29 | 68.12 | 42.30 | 63.60 | 54.90 | 61.44 |
| Phi3-3B | | | | | | |
| Baseline | 62.98 | 53.00 | 33.88 | 56.89 | 42.70 | 49.89 |
| ProxyV - Layer 16 | 62.55 | 53.72 | 34.67 | 56.98 | 43.50 | 50.28 |
| ProxyV - Layer 20 | 64.78 | 54.04 | 33.60 | 56.48 | 44.70 | 50.72 |
| InternLM2.5-7B | | | | | | |
| Baseline | 72.46 | 64.80 | 38.38 | 63.40 | 52.60 | 58.33 |
| ProxyV - Layer 16 | 74.00 | 65.20 | 39.41 | 62.70 | 52.10 | 58.68 |
| ProxyV - Layer 20 | 74.24 | 66.04 | 38.66 | 63.98 | 52.50 | 59.08 |

