# OpenReview forum: "Streamline Without Sacrifice - Squeeze out Computation Redundancy in LMM"
_ICML.cc/2025/Conference — ICML 2025 poster_

### Official Review · Reviewer_iBst · 2025-02-16

**Overall Recommendation:** 4

**Summary:**

This paper focuses on the computational redundancy of visual tokens in large multimodal models (LMMs). It is found that there is computational-level redundancy in visual tokens within LMMs, and different LLMs exhibit varying degrees of redundancy. The ProxyV algorithm is proposed. By introducing proxy visual tokens, it alleviates the computational burden of original visual tokens, improves efficiency without sacrificing performance, and can even enhance performance in some scenarios. Experiments verify the effectiveness of ProxyV on different LLMs. Moreover, a non-spatial variant is designed, which can be combined with token reduction methods to further boost efficiency.

**Claims And Evidence:**

The evidence is partially insufficient. Although a series of experiments in the paper verify the existence of computational redundancy and the effectiveness of ProxyV, the evidence for some key conclusions is incomplete. For example, when analyzing the redundancy patterns of visual attention in different LLMs, only the experimental results of a limited number of models (such as Vicuna1.5-7B, Qwen2-7B, etc.) are presented, which is difficult to represent all LLMs. We would like to see results on larger models (32B).
The analysis of the reasons for performance improvement is inadequate. The ProxyV algorithm brings performance improvement. The paper attributes it to the decoupled vision-specific modules. However, there is a lack of in-depth analysis and evidence on how these modules specifically act on different tasks and model structures, making it difficult for readers to clearly understand the internal mechanism of performance improvement.

**Essential References Not Discussed:**

No more referecens needs to be discurssed.

**Experimental Designs Or Analyses:**

Comprehensiveness of experimental design: The overall experimental design is relatively comprehensive. By comparing different experimental settings (such as different attention masking positions and different model structures), the computational redundancy problem and the performance of the ProxyV algorithm are systematically studied. However, in the comparison experiments, for some experiments combining with other methods (such as the combination of non-spatial ProxyV and VisionZip), the experimental design could be more detailed. For example, experimental results under different combination ratios could be added to more comprehensively evaluate the combination effect.
Depth of experimental analysis: The experimental analysis mainly focuses on the comparison of performance indicators (such as Score, FLOPs, Time). The analysis of the reasons behind the experimental results is not in-depth enough. For example, when ProxyV performs differently in different tasks (fine-grained and coarse-grained tasks), only the early or late appearance of computational redundancy is simply mentioned, and the internal relationship between task characteristics and algorithm performance is not deeply explored.

**Methods And Evaluation Criteria:**

Rationality of the method: The ProxyV algorithm, aiming at the computational redundancy of visual tokens, uses proxy tokens. Theoretically, it can effectively reduce the computational load. The method is reasonably designed. Compared with traditional token reduction methods, it avoids information loss and is innovative.
Evaluation criteria: The selected OCR-extensive benchmark tests and other related datasets, such as DocVQA and ChartQA, are suitable for evaluating the performance of models in processing visual information tasks. They can effectively test the model's ability to understand and process fine-grained visual information. However, for more complex tasks in real-world scenarios, such as dynamic scenarios of multimodal information fusion, the evaluation criteria may not be comprehensive enough.

**Other Comments Or Suggestions:**

Supplement more theoretical analysis.
Validate on more benchmarks.

**Other Strengths And Weaknesses:**

The results of applying ProxyV to different layers could be presented with more fine-grained details.

**Questions For Authors:**

In the experiments, only a limited number of LLMs were tested. How can you ensure that the ProxyV algorithm is equally effective on larger-scale LLMs (32B)? And what is the performance under larger data volumes (such as Cambrian-10M or Llava-onevision)? If you can provide experimental results or theoretical analysis on more models, it will enhance the universality of the paper's conclusions, and my evaluation of the paper will be more positive.
The results of ATN+FFN in the paper seem to be good enough. Please compare the experiments of ATN+FFN and ProxyV in detail under fair settings.
In Figure 2, the results of ATN+FFN are better than those of ATN, and it achieves better performance with less computational effort. Please analyze the reasons.

**Relation To Broader Scientific Literature:**

Relationship with existing models: The work of this paper is closely related to current mainstream research on large multimodal models. In terms of improving computational efficiency, it contrasts with traditional token reduction methods, pointing out that token reduction methods have the problem of information loss, while the ProxyV algorithm of this paper avoids this problem by reducing computational redundancy. Compared with cross-attention-based LMMs, ProxyV reduces computational costs while maintaining the simplicity and efficiency of the decoder structure.
Expansion of research direction: Based on the existing research's focus on visual information processing and computational efficiency, this paper further explores the computational redundancy of visual tokens, providing new ideas and methods for improving the efficiency of large multimodal models and expanding the research direction in this field.

**Theoretical Claims:**

There is no theoretical content.

---

> ### Author Rebuttal · Authors · 2025-03-31
>
> We appreciate your constructive and insightful feedback. We carefully considered your reviews and revised paper accordingly. Below, we directly address each point you raised:
> - *Incomplete evidence for the existence of computational redundancy*
>   Thanks for your suggestion to validate our observations on larger-scale models. Initially, we explored redundancy patterns using six widely used LLMs to broadly cover commonly studied architectures. Following your recommendation, we additionally conducted experiments on a larger-scale model, Qwen-2.5-32B (using LLaVA-1.5 style image encoding), and observed similar computational redundancy patterns (See updated [Figure here](https://anonymous.4open.science/r/ICML2025_Paper2984_Rebuttal_Figures-B873/Figure2_revised.png)). This confirms that our earlier findings generalize to larger models.
> - *Lack of in-depth analysis for the performance improvement*
>    Your suggestion to clarify the internal mechanisms underlying performance improvements is very valuable. To elucidate this, we measured the MIR (Modality Integration Rate) score [1], which quantifies alignment between visual and textual tokens (lower scores indicate better alignment). Evaluating 100 randomly sampled instances from the DetailCaps-4870 benchmark [2], we found that applying ProxyV-L12 to Vicuna-7B decreased the MIR score from 3.62 to 3.10, reflecting improved modality alignment due to decoupled vision modules. We will expand upon this insightful analysis in our revised manuscript.
> - *Internal relationship between task characteristics and algorithm performance*
> Thanks for your constructive suggestion. Instead of defining tasks as fine-grained and coarse-grained based on human priors, we will provide quantitative metrics to represent task characteristics. Here we take ChartQA and GQA as two representative tasks, and we downscale the input image to 336*336 and measure the relative performance drop. ChartQA's performance drops by 28.7%, while GQA's performance drops by only 0.1%, indicating ChartQA requires much more fine-grained information than GQA. Then, we conduct experiments with the ATN+FFN (skipping both visual attention and FFN) variant to study the appearance of visual computation redundacy. We find that to retain the original performance, the skipping operation has to be applied after layer-17 for ChartQA and only layer-6 for GQA. This indicates that the visual computation redundancy appears from early layers on less fine-grained dataset GQA. We will thoroughly detail these results and analyses in the revised paper.
> - *Evaluation criteria may not be comprehensive enough*
>   Following your advice, we expanded our evaluations across additional MLLM benchmarks (see our response to Reviewer nPTg). We will provide results for all model variations in our revision.
> -  *The results of applying ProxyV to different layers could be presented with more fine-grained details.*
> We provide the full evaluation results on each benchmark in Section A of the Supplementary.
> - *Validate ProxyV with larger scale data & models*
> Thanks for your valuable advice. We further increase the data scale to 3M and validate our method with Vicuna-7B (results shown in the table below) and observe a consistent conclusion. Due to computational constraints, we currently do not have results for even larger datasets or models such as 32B-scale models, but we plan to explore this in future work.
>
> |SFT Data|**Avg**|DocVQA|ChartQA|InfoVQA|TextVQA|OCRBench|
> |---------------|---------|---------|----------|---------|--------|-------|
> |Baseline (779K)|**54.64**|68.03|59.64|33.60|62.12|49.80|
> |ProxyV-L16 (779K)|**55.94**|69.90|61.48|34.24|62.28|51.80|
> |Baseline (3M)|**64.89**|79.45|65.32|48.73|67.84|63.1|
> |ProxyV-L16 (3M)|**65.97**|80.3|67.04|50.20|69.01|63.3||
>
> - *Results of ATN+FFN*
>   In Table 2, the performance of ATN+FFN is much worse than ATN for the layer-0 case, achieves similar performance for the layer-12 case, and improves the performance for the layer-16 case. We have analyzed the reason in the last part of Section 2 and attribute the performance gain to the additional vision specific modules in ATN+FFN and the negative impact of additionally skipping FFNs becomes smaller in later layers. Our focus is to ensure no performance drop after the acceleration. Note that ATN+FFN still incurs performance degradion when applying from Layer 12 ($Score_{fine}$ = 54.28) and only achieves no performance loss applying from ($Score_{fine}$ = 54.85), while ProxyV-L12 achieves performance gain ($Score_{fine}$ = 55.16). In this case, ProxyV-L12 also has better efficiency than ATN-FFN-L16. All experiments are conducted in the same setting for fair comparison.
>
> [1] Huang, Qidong, et al. "Deciphering Cross-Modal Alignment in Large Vision-Language Models with Modality Integration Rate." arXiv preprint arXiv:2410.07167 (2024).
>
> [2] Dong, Hongyuan, et al. "Benchmarking and improving detail image caption." arXiv preprint arXiv:2405.19092 (2024).

---

> > ### Comment · Reviewer_iBst · 2025-04-06
> >
> > Thanks for the authors' feedback. Your feedback address my concerns. I am interested to see your method intergrated with hybrid LLMs like jamba or samba. Does your method is general to be integrated with these efficient hybrid stcutuctures?

---

> > > ### Author Response · Authors · 2025-04-06
> > >
> > > Thank you for your kind follow-up and positive feedback.
> > > Yes, our method is designed to be general, and it can be integrated with hybrid architectures to further reduce computation on vision tokens in MLPs and (window) attention layers.
> > > We are excited about this direction and plan to explore ProxyV’s integration with hybrid LMMs in future work.

---

### Official Review · Reviewer_nPTg · 2025-03-13

**Overall Recommendation:** 1

**Summary:**

This paper focuses on the token acceleration of LMM. It understands the hierarchical redundancy on visual tokens through validation experiments, and finds that visual tokens from LMM visual encoders do not necessarily require all heavy operations in the decoder-only LMM. ProxyV is designed to reduce computational burden with slight loss of information, on some VQA benchmarks.

**Claims And Evidence:**

This manuscript was inspired by experiments to develop specific designs, thus many priors may not necessarily have generalizability. For example, in Line 106-129, "directly masking vision token attention across the entire LMM leads to a significant performance drop while masking it from the middle or later layer has minimal or no effect on performance." Does the network only need to be divided into three levels? If the network is deeper, does the conclusion still hold?

**Essential References Not Discussed:**

There are several methods of token acceleration are missing, which have proposing some similar experimental findings.

[1] Compression with Global Guidance: Towards Training-free High-Resolution MLLMs Acceleration. ArXiv 2024

[2] Rethinking Token Reduction in MLLMs: Towards a Unified Paradigm for Training-Free Acceleration. ArXiv 2024

[3] FOLDER: Accelerating Multi-modal Large Language Models with Enhanced Performance. ArXiv 2024

**Experimental Designs Or Analyses:**

After reviewing all the experimental analyses, the reviewer found many results of this paper somewhat unconvincing. For reduced FLOPs/Time, there is no specific value, only the change proportion (In fact, there is a lot of blank space in the manuscript). For comparisons with existing methods, there are only two competitors (and not SOTA).

**Methods And Evaluation Criteria:**

Overall, the evaluation criteria and benchmark are insufficient, which makes the work's contribution to the community somewhat minor. As LMM has wide application scenarios, only using DocVQA\ChartQA\InfoVQA\TextVQA\OCRBench for evaluation is not convincing enough.
n fact, if evaluated on more comprehensive LMM benchmarks (such as fine-grained vision, retrieval, detection, and segmentation), it would be better to assess the multi-modal understanding and reasoning capability.

**Other Comments Or Suggestions:**

Please consider the font size and information content of the table/image. Most figures are difficult to see in terms of text size, and most tables have results that are not detailed enough.

**Other Strengths And Weaknesses:**

[-] Insufficient Comparisons. This paper only compares with VisionZIP and PyramidDrop, which is unconvincing. And in terms of benchmarks, there are only so-called fine-grained VQA, without comprehensive evaluation for the LMM's performance.

[-] Unclear Details. Many of results and validation experiments do not provide specific details, which makes the reviewer a bit confused about the validity of the conclusions.

[-] Unfair Comparisons. To squeeze out computation redundancy in LMM, this paper introduces ProxyV tokens with additional costs. How does Tab. 4-11 consider these additional costs?

**Questions For Authors:**

Please see the weaknesses.

**Relation To Broader Scientific Literature:**

Compared to mainstream token compression ideas, this paper focuses on using some proxy tokens to compensate for performance, which makes it somewhat similar to one prompt-tuning approach (learnable tokens) applied to the acceleration of LMM.

[1]  Learning to prompt for vision-language models. IJCV 2022

[2] Prefix-tuning: Optimizing continuous prompts for generation. ACL 2021

**Theoretical Claims:**

The paper does not propose any theoretical claims.

---

> ### Author Rebuttal · Authors · 2025-03-31
>
> Thank you very much for your thoughtful feedback. We have carefully considered your comments and made detailed revisions to address each point as follows:
> - *Generalizability of Claims*
> The exploratory experiments presented in Section 2 are designed to identify computational redundancy patterns rather than propose universal claims or laws. Through our experiments on a wide range of LLMs, we can observe clear signals of computational redundancy on visual tokens, and we also explicitly state that "different models exhibit different patterns" in L130. Our intent was to highlight potential redundancy, motivating further focused studies. We also additionally add experiments on a deeper model, Qwen-2.5-32B (using LLaVA-1.5 style image encoding), and observed similar computational redundancy patterns (See the updated [Figure here](https://anonymous.4open.science/r/ICML2025_Paper2984_Rebuttal_Figures-B873/Figure2_revised.png)). We will revise our paper to make the claims more clear.
> - *Comprehensive Benchmarking and Evaluation*
> We appreciate your suggestion regarding the comprehensiveness of evaluation benchmarks. Accordingly, we have expanded our evaluation to include multiple popular MLLM benchmarks. Our extended evaluation (detailed below) clearly shows that our method consistently achieves equal or superior performance compared to the baseline, while other token reduction methods show a notable gap in **fine-grained visual grounding tasks like RefCOCO**. We will provide additional comprehensive evaluations of all LLM variants in our next revision. Also, we clarify that we initially mainly focused on fine-grained benchmarks as they are good indicators for possible visual information loss, which is critical for MLLM acceleration methods.
>
> | Model Group| Model| Avg| MMBench |SEED-Img|RefCOCO|MMStar|GQA|MME-P|MMMU|POPE|SQA|AI2D|RealWorldQA|
> |---------------|---------------|---------|---------|----------|---------|--------|-------|---------|--------|-------|--------|-------|--------------|
> | **Vicuna-7B** | Baseline      | 63.35 | 65.03   | 68.60    | 75.39   | 37.70  | 63.36 | 1428.52 | 36.56  | 86.91 | 68.52 | 66.71 | 56.73        |
> | |VisionZip| 63.27 |65.37|68.74|71.63| 37.38|63.93|1437.50| 34.33  | 87.44 | 69.51 | 68.30| 57.52  |
> ||Pdrop| 63.24 |65.37| 68.45|72.53|36.66|63.84| 1451.33|35.56|87.10|67.63|67.94|58.04|
> ||**ProxyV**|**64.44**| 67.61|70.02|76.76|38.35| 64.02 | 1478.61 | 35.44  | 86.55 | 68.07 | 69.11 | 59.08 |
> |**Vicuna-13B**| Baseline|66.23|67.78|70.45| 81.69   | 41.60  | 65.19 | 1617.57 | 34.56  | 86.58 |71.05 |69.88 | 58.95 |
> |               | **ProxyV**         | **67.20** | 69.15   | 71.71| 83.30|42.40 |65.59|1602.15|36.00 | 87.16 | 73.53 | 71.50 | 58.82        |
> - *Unclear experimental details*
> We have provided our experimental setting and details (data, training pipeline, model structure, image encoding scheme) for all experiments in Section 4 and provide more model hyperparameter details in Section B of the Supplementary. We will add more experimental and evaluation details in our revised version.
> - *Absolute values for FLOPs & Time, ProxyV’s additional costs*
> Thank you for your valuable suggestion. We provide the absolute FLOPs & time values for Vicuna-7B/13B based models below, and we will add the results for all models in our revised version. All the FLOPs and times are directly measured in the same setting for all models, so the additional operations in ProxyV are already included. We will add clarification about this in the paper.
>
> | Model Group   | Model         | Time (s)     | FLOPS (T) |
> |---------------|---------------|---------|---------|
> | **Vicuna-7B** | Baseline      | 0.252 | 42.47   |
> |               | ProxyV-L12     | 0.148 | 23.13   |
> |               | ProxyV-L16     | 0.173 | 27.00   |
> | **Vicuna-13B**| Baseline      | 0.411 | 81.42    |
> |               | ProxyV-L16        | 0.244 | 46.01   |
> |               | ProxyV-L20        | 0.274 | 51.91   |
> - *Missing Essential References & Only compared with two non-SOTA competitors*
> Thank you for recommending additional related papers. According to the ICML 2025 review instructions, "Authors cannot expect to discuss other papers that have only been made publicly available within **four months** of the submission deadline." PyramidDrop (CVPR 2025) is the most recent token reduction method that has been peer-reviewed to the best of our knowledge and can well represent the SOTA performance of token reduction methods. Besides, we emphasize that our proposed ProxyV approach is orthogonal and complementary to token reduction methods, with combined effectiveness demonstrated in Section 3.3.
> - *Figure font size and detailed results in tables*
> We appreciate your suggestion on improving readability. We have modified our figures in this [link](https://anonymous.4open.science/r/ICML2025_Paper2984_Rebuttal_Figures-B873). The detailed benchmark results for all experiments are provided in Section A of the Supplementary.

---

> > ### Comment · Reviewer_nPTg · 2025-04-09
> >
> > Thanks for your efforts during rebuttal. I have carefully read these additional experiments, a few concerns about absolute values for FLOPs & Time, ProxyV’s additional costs have been addressed.
> >
> >
> > However, most concerns about differences from the idea of applying prompt-tuning to LMM’s acceleration, unclear details such as hyper-parameters and robustness, comprehensive benchmarking and evaluation are still exist. Thus, I tend to maintain the rating.

---

> > > ### Author Response · Authors · 2025-04-09
> > >
> > > Thank you for your feedback. We would like to clarify that the prompt-tuning methods you mentioned are parameter-efficient fine-tuning techniques, which are fundamentally different from the goal and design of current LMM acceleration methods.
> > >
> > > Additionally, we have provided evaluation results on 11 widely used LMM benchmarks in the rebuttal to comprehensively validate the effectiveness and robustness of our approach. The experimental settings, including hyper-parameters, are detailed in Section 4 of the main paper and Section B of the Supplementary.

---

### Official Review · Reviewer_JDVo · 2025-03-14

**Overall Recommendation:** 2

**Summary:**

Summary:
This paper explores the computational redundancy inherent in vision tokens within multimodal large language models. This paper reveals significant computational redundancy exists in vision token processing, particularly in the middle and later layers of such models. To address this inefficiency, this paper proposes ProxyV, a lightweight approach that optimizes vision token computation by introducing compressed proxy tokens and efficient update mechanisms.

Strength:
1.	This paper explores computational redundancy in vision tokens through extensive experiments.
2.	The proposed ProxyV method successfully reduces the computation overhead and inference time.

Weakness:
1.	The article needs to be better organized. The font in the figure is too small and difficult to read.
2.	The article lacks formal expression and the details are difficult to confirm.
3.	The structure of the article needs further organization. The author gives a lot of experimental results in the method section, but does not explain the specific experimental settings.
4.	The author's experimental and comparison methods should be aligned with [1][2]. The current experimental results lack comparison on a wide range of MLLM benchmarks.

[1] PyramidDrop: Accelerating Your Large Vision-Language Models via Pyramid Visual Redundancy Reduction
[2] VisionZip: Longer is Better but Not Necessary in Vision Language Models

**Claims And Evidence:**

See summary

**Essential References Not Discussed:**

See summary

**Experimental Designs Or Analyses:**

See summary

**Methods And Evaluation Criteria:**

See summary

**Other Comments Or Suggestions:**

See summary

**Other Strengths And Weaknesses:**

See summary

**Questions For Authors:**

See summary

**Relation To Broader Scientific Literature:**

No

**Theoretical Claims:**

See summary

---

> ### Author Rebuttal · Authors · 2025-03-31
>
> Thank you for your valuable feedback. We have reviewed your suggestions thoroughly and made corresponding revisions to address each of your concerns as outlined below.
> - *Paper Organization and Readability*
>   We appreciate your suggestion on improving readability. We have modified our figures, and the revised figures can be found in this [link](https://anonymous.4open.science/r/ICML2025_Paper2984_Rebuttal_Figures-B873). We will reorganize the paper for clarity. We will add clearer captions, make expressions more consistent, and provide a more formal, detailed explanation of our approach to ensure better understanding.
> - *Experimental Details*
>   We have provided our experimental setting and details (data, training pipeline, model structure, image encoding scheme) for all experiments in Section 4 and provide more model hyperparameter details in Section B of the Supplementary. We will add more experimental and evaluation details in our revised version.
> - *Comparison on more MLLM benchmarks*
>   In response to your valuable comment on broader comparisons, we have expanded our evaluation to include additional standard benchmarks for MLLMs, as summarized in the table below. Our revised manuscript will include extensive evaluations across different LLM variants. From this table, we can see our method ensures no performance loss or achieves performance gain on most benchmarks, while token reduction methods have inferior performance on grounding benchmarks like RefCOCO, which requires finer visual information. Also, we clarify that we initially mainly focused on fine-grained benchmarks as they are good indicators for possible visual information loss, which is critical for MLLM acceleration methods.
>
> | Model Group   | Model         | Avg     | MMBench | SEED-Img | RefCOCO | MMStar | GQA   | MME-P   | MMMU  | POPE  | SQA   | AI2D  | RealWorldQA |
> |---------------|---------------|---------|---------|----------|---------|--------|-------|---------|--------|-------|--------|-------|--------------|
> | **Vicuna-7B** | Baseline      | 63.35 | 65.03   | 68.60    | 75.39   | 37.70  | 63.36 | 1428.52 | 36.56  | 86.91 | 68.52 | 66.71 | 56.73        |
> |               | VisionZip     | 63.27 | 65.37   | 68.74    | 71.63   | 37.38  | 63.93 | 1437.50 | 34.33  | 87.44 | 69.51 | 68.30 | 57.52        |
> |               | Pdrop         | 63.24 | 65.37   | 68.45    | 72.53   | 36.66  | 63.84 | 1451.33 | 35.56  | 87.10 | 67.63 | 67.94 | 58.04        |
> |               | **ProxyV**         | **64.44** | 67.61   | 70.02    | 76.76   | 38.35  | 64.02 | 1478.61 | 35.44  | 86.55 | 68.07 | 69.11 | 59.08        |
> | **Vicuna-13B**| Baseline      | 66.23 | 67.78   | 70.45    | 81.69   | 41.60  | 65.19 | 1617.57 | 34.56  | 86.58 | 71.05 | 69.88 | 58.95        |
> |               | **ProxyV**         | **67.20** | 69.15   | 71.71    | 83.30   | 42.40  | 65.59 | 1602.15 | 36.00  | 87.16 | 73.53 | 71.50 | 58.82        |

---

### Decision · Program_Chairs · 2025-05-01

**Decision:**

Accept (poster)

**Comment:**

This paper presents an approach for token pruning that introduces a few proxy visual tokens to reduce the burden from the original tokens.  The authors provide both a study of the redundancy pattern of tokens, motivating the proposed approach.  The majority of reviewers commended rejecting the paper in its current form.  However, one reviewer did not update their review post-rebuttal, and the ACs feel their comments were addressed.  The other reviewer requested experiments comparing to another task, prompt-tuning, which the reviewer argues is conceptually similar.  While the AC agrees that there can be parallels drawn between these approaches, the goals of the two tasks are different.  Specifically, prompt-tuning typically aims to reduce training resources (typically without evaluating test-time inference time), whereas token pruning focuses on test-time inference.  Thus, even if they are conceptually similar, it does not diminish their contributions as the two tasks have wildly different applications.  The AC does agree with the comment made by all reviewers that the experimental details are lacking, especially the in-depth comparisons to prior work (e.g., comparing against methods like PruMerge or FastV, even if they are expected to perform worse than other baselines).  The results in the rebuttal only partly address these concerns, and the authors are encouraged to provide several computational budgets on a similarly rigorous evaluation comparing against prior work in their next version.